# Understanding the psychological mechanisms of return to sports readiness after anterior cruciate ligament reconstruction

**Bernard X. W. Liew**[1]*, **Julian A. Feller**[2], **Kate E. Webster**[3]

**1** School of Sport, Rehabilitation and Exercise Sciences, University of Essex, Colchester, United Kingdom,
**2** Ortho Sport Victoria, Epworth Health Care, Melbourne, Victoria, Australia, **3** School of Allied Health,
Human Services and Sport, La Trobe University, Melbourne, Victoria, Australia

* bl19622@essex.ac.uk, liew_xwb@hotmail.com

## Abstract

### Purpose

The psychological response to an Anterior Cruciate Ligament (ACL) injury is significant and can negatively impact return to sports outcomes. This study aimed to quantify the association between factors associated with return to sport using network analysis.

### Methods

441 participants who underwent primary ACL reconstruction. The 12-item ACL Return to Sport after Injury (ACL-RSI) scale was administered to all participants 12 months after surgery. Three network analyses were used to quantify the adjusted correlations between the 12 items of the ACL-RSI scale, and to determine the centrality indices of each item (i.e., the degree of connection with other items in the network). Further subgroup network analyses were conducted for those who had (n = 115) and had not returned (n = 326) to their pre-injury level of sport.

### Results

The greatest adjusted correlation was between Q7 and Q9 (fear of re-injury and afraid of accidentally injuring knee) of the ACL-RSI (group 0.48 (95%CI [0.40 to 0.57])) across all three networks. The most important item in the network was Q12 (relaxed about sport) across all three networks. Individuals who did return to sport had greater Strength centrality for Q8 (confidence in knee, P = 0.014) compared to those who did not return to sport.

### Conclusion

Fear of re-injury and being relaxed about playing sport were the two most important nodes in the network models that describe the return to sport readiness. The importance of knee confidence at influencing psychological readiness was greater in athletes who did return to sport compared to those who did not. Our findings provide candidate therapeutic targets

**Data Availability Statement:** All relevant data are available through Zenodo at DOI: 10.5281/zenodo.6339411.

**Funding:** The author(s) received no specific funding for this work.

**Competing interests:** The authors have declared that no competing interests exist.

that could inform future interventions designed to optimize return to sport rates in athletes post ACL reconstruction.

## Introduction

An anterior cruciate ligament (ACL) rupture is a serious knee injury that usually occurs during sports participation [1]. It is typically treated with surgical reconstruction and most athletes aim to resume their pre-injury levels of sports participation [2]. Much research has examined how knee function is affected by this surgery and what deficits persist even after rehabilitation programs are complete [3,4]. Although less attention has been paid to the psychological consequences of ACL injury, it is now well recognised that the psychological response to this injury is significant and continues long after the injury has occurred [5–10]. The psychological response to an ACL injury can therefore have a negative impact on rehabilitation and return to sports outcomes [6–8].

The ACL Return to Sport after Injury (ACL-RSI) scale aims to measure psychological readiness to return to sport following ACL injury or surgery [11]. It is the only psychological scale specific to ACL injury and significant validation work has been undertaken [11,12]. The ACL-RSI is composed of 12 items and the total score indicates the overall psychological readiness level for a return to sport after the injury. Items in the scale are centred on three psychological factors: *emotions*, *confidence in performance*, and *risk appraisal* [11]. A fundamental theoretical construct underpinning the contemporary use and interpretation of the ACL-RSI is known as the "reflective model" (RM) [13]. Put simply, observed item responses on the ACL-RSI are determined by a latent trait—readiness. The advantage of using the total score is that it makes it easier for performing traditional statistical modelling.

The total score of the ACL-RSI may have disadvantages. First, two individuals could have identical ACL-RSI total scores but with different item responses. Understanding what precisely is being affected in people with an ACL injury is required for providing individualised treatment. Second, the relationship between different items of the ACL-RSI, a feature not captured when using only an aggregate score, may be just as important as individual item responses in providing a holistic understanding of the state of readiness in individuals with an ACL injury [14]. This would mean that simultaneous changes to the responses of multiple items, thereby influencing their relationship, may be important in influencing psychological readiness recovery in individuals with an ACL injury.

Qualitative studies have supported the notion that psychological factors associated with sports injury are a dynamic and complex construct [15]. A quantitative method to measure such complexity in ACL injured participants is network analysis [16]. In network analysis, individual ACL-RSI items are treated as *nodes*, associations between two nodes in a network are connected by an "edge", and a network model conceptualizes readiness as a set of mutually interacting associations between these *nodes*. In addition, the number and strength of associations between a node and all other nodes indicate the node's relative importance within the network. Such information could be used for guiding future interventions [17]. Previous studies have reported that fear of movement [18,19] and knee confidence [18] are important determinants of return to sport. Whether nodes that reflect fear and confidence are likely to demonstrate greater importance within a network model in those that did return to sport, compared to does that did not return have not been determined.

To our knowledge, network analysis has not featured in ACL research but has been used in general psychological disorders [20–22]. A previous study reported that athletes tend to score

lower on the emotion-based items (i.e., have more of a negative emotional response) than for the confidence or risk appraisal items [11]. Hence, we hypothesized that items within each of three psychological factors (*emotions*, *confidence*, and *risk*) would exhibit greater association, than compared to items reflecting different factors. In addition, we also hypothesized that emotion-based items will have greater measures of importance within the network than confidence or risk appraisal items. Lastly, we explored the hypothesis that nodes that reflect fear and confidence are likely to demonstrate greater importance within the network in those that did return to sport, compared to those that did not return.

## Methods

### Participants

The study consisted of 441 participants (184 female, 257 male), who had undergone primary ACL reconstruction surgery. Participants were eligible for inclusion if they had played sport (minimum 1–3 days per month) before the ACL injury, had no prior contralateral ACL injury, and attended a scheduled 12-month review appointment following surgery. All had undergone arthroscopically assisted surgery and no other ligament damage was present. Rehabilitation protocols and guidelines were provided which encouraged immediate full knee extension and the restoration of quadriceps function as soon as possible [23]. Beginning at 3 weeks, stationary bike, wall squats, straight-leg raises, forward lunges, and hamstring curls were introduced. At 5 weeks, a gymnasium program commenced that included leg press, half squats, stationary bike, rowing machine, cross-trainer and step-machine, hamstring curls, calf raises, exercise ball drills for core stability, and leg extensions (after 8 weeks). At 10 weeks, hopping and landing drills were commenced if there was no effusion. At 16 weeks, patients were typically allowed to return to sport-specific drills and activities. At 26 weeks, patients were encouraged to increase training intensity [23]. Clearance to return to competitive sport was typically between 9–12 months post-surgery and was determined by the treating surgeon. Exclusion criteria were any further surgery or subsequent ACL injury during the follow-up period. Ethical approval was granted from hospital and university ethics committees. Written informed consent was sort from all participants prior to study enrolment.

### Study design

The present analysis was undertaken on a prospective cohort data set that was part of a larger study on ACL outcomes at one institution. Participants were enrolled in this study before undergoing ACL reconstruction surgery. They were scheduled for routine post-surgical follow-up, which included an assessment at 12 months which was the time point used in the current analysis. This time point was chosen as changes in psychological readiness to return to sport are expected at this time as training and return to play recommences.

### Approach to network analysis

The ACL-RSI was gathered from all participants at 12-month follow-up post-surgery. Network analysis was performed for the entire cohort (n = 441), and subgroup analyses were performed in those who did (n = 115) and did not return (n = 326) to pre-injury level sports at 12 months. Participants who self-reported "Yes, at the same or higher level compared to before injury" were classified as a return to sport; whilst those who self-reported "No", "Yes, training only", or "Yes, at a lower level compared to before injury" were classified as not returning to pre-injury level sports.

**Software and packages.** The data set was analysed with the R software for statistical computing (version 4.1.2), and can be found (https://zenodo.org/record/6339411#.YifO_XrP2Uk). Several packages were used to carry out the analyses, including *qgraph* for network estimation [24], *bootnet* for stability analysis [25], and *NetworkComparisonTest* for network comparison [26].

**Variables included in network analysis.** A network structure is composed of nodes and edges. In our study, the 12 items of the ACL-RSI [11] were used as nodes and were included in the network model as continuous variables (Table 1). Edges represent an association between two nodes, adjusted for all other nodes. Each edge in the network represents either positive regularized associations (blue edges) or negative regularized associations (red edges). The thickness and colour saturation of an edge denotes its weight (the strength of the association between two nodes).

A nonparanormal transformation was applied to ensure that these 12 variables were multivariate normally distributed [27]. When estimating the network, a form of least absolute shrinkage and selection operator (LASSO) regularization [28], termed graphical LASSO [29], which utilizes penalized maximum-likelihood estimate, was used to elicit a sparse model. The LASSO uses a tuning parameter to control the sparsity of the network, which we chose by minimizing the Extended Bayesian Information Criterion (EBIC) [30].

**Node centrality.** Centrality indices provide a measure of a node's importance, and they are based on the pattern of connectivity of a node of interest with its surrounding nodes. In the present study, we calculated the Strength centrality, which is defined as the sum of the weights of the edges (in absolute value) incident to the node of interest [31,32]. Clinically, a high Strength node represents potentially good therapeutic targets, because a change in the

**Table 1. Anterior cruciate ligament return to sport after injury scale.**

| Variables | Question | Scale (0–100) numerical rating scale | Attribute |
|---|---|---|---|
| Q1 | Are you confident that you can perform at your previous level of sport participation? | 0 = Not at all confident 100 = Fully confident | Confidence in performance |
| Q2 | Do you think you are likely to re-injure your knee by participating in your sport? | 0 = Extremely likely 100 = Not likely at all | Risk appraisal |
| Q3 | Are you nervous about playing your sport? | 0 = Extremely nervous 100 = Not nervous at all | Emotion |
| Q4 | Are you confident that your knee will not give way by playing your sport? | 0 = Not at all confident 100 = Fully confident | Confidence in performance |
| Q5 | Are you confident that you could play your sport without concern for your knee? | 0 = Not at all confident 100 = Fully confident | Confidence in performance |
| Q6 | Do you find it frustrating to have to consider your knee with respect to your sport? | 0 = Extremely frustrating 100 = Not at all frustrating | Emotion |
| Q7 | Are you fearful of re-injuring your knee by playing your sport? | 0 = Extremely fearful 100 = No fear at all | Emotion |
| Q8 | Are you confident about your knee holding up under pressure? | 0 = Not at all confident 100 = Fully confident | Confidence in performance |
| Q9 | Are you afraid of accidentally injuring your knee by playing your sport? | 0 = Extremely afraid 100 = Not at all afraid | Emotion |
| Q10 | Do thoughts of having to go through surgery and rehabilitation again prevent you from playing your sport? | 0 = All of the time 100 = None of the time | Risk appraisal |
| Q11 | Are you confident about your ability to perform well at your sport? | 0 = Not at all confident 100 = Fully confident | Confidence in performance |
| Q12 | Do you feel relaxed about playing your sport? | 0 = Not at all relaxed 100 = Fully relaxed | Emotion |

value of this node has a strong direct, and quick (because of its strong direct connections), influence on the nodes within the network.

**Accuracy and stability.** We assessed the accuracy of the edge weights and the stability of three centrality indices using bootstrapping [25]. We bootstrapped using 1000 iterations and report the 95% confidence intervals (CI) of all edge weights. To gain an estimate of the variability of Strength centrality, we applied the case-dropping subset bootstrap [25]. This procedure drops a percentage of participants, re-estimates the network, and re-calculates the centrality index; producing a centrality-stability coefficient (CS-coefficient). CS reflects the maximum proportion of cases that can be dropped, such that with 95% probability the correlation between the centrality value of the bootstrapped sample vs that of the original data, would reach a certain value, taken to be a correlation magnitude of 0.7 presently. It is suggested that $CS_{cor\ =\ 0.7}$ should not be below 0.25 and better if $> 0.5$ [25].

**Network comparison.** Between-group (returners vs non-returners) comparisons of pairwise node associations and Strength index of each node were computed using the network comparison tests (NCT) [26]. The NCT is a 2-tailed permutation test in which the difference between two groups (those who did and did not return to sports) is calculated repeatedly (1000 times) for randomly sorted participants. This results in a null hypothesis distribution (assuming that both groups are equal), which can be used to test the observed difference between the groups. A previous study used a threshold of 0.05 to determine statistical significance during NCT [33]. Given the exploratory nature of the present study, we did not adjust this threshold for multiple comparisons. Hence, we considered between-group differences with a P value $< 0.05$ as having more evidence in favour of the alternative hypothesis, than differences with a P value $\geq 0.05$.

## Results

Table 2 represents the baseline characteristics of the participants. The mean (standard deviation [SD]) of the variables (original scale) used in the network analysis can be found in the Table 3. Fig 1 shows the networks of the entire cohort at 12-months follow-up and the subgroups of those who did or did not return to sports at this time point.

**Table 2. Baseline participant characteristics.**

| Variables | Values |
|---|---|
| Age at surgery (years)* | 24.6(7.4) |
| Sex | |
| Female | 184(41.7) |
| Male | 257(58.3) |
| Sporting level prior to injury | |
| Professional | 10 (2) |
| High-level competition sport | 169(38) |
| Frequent sport | 232 (53) |
| Sport sometimes | 30 (7) |
| Frequency | |
| 4–7 days/week | 254 (58) |
| 1–3 days/week | 176 (40) |
| 1–3 times/month | 11 (2) |

* Values represent mean (standard deviation).

Categorical variable values represent count (percentage).

**Table 3. Anterior cruciate ligament return to sport after injury individual item mean (1 standard deviation) scores.**

| Items | Whole cohort (n = 441) | Did not return to sport (n = 326) | Return to sport (n = 115) |
|---|---|---|---|
| Q1 | 80.78 (22.94) | 71.74 (26.36) | 89.3 (14.8) |
| Q2 | 67.34 (26.02) | 59.82 (25.49) | 74.44 (24.53) |
| Q3 | 59.34 (31.15) | 47.92 (28.53) | 70.1 (29.69) |
| Q4 | 74.15 (25.68) | 63.77 (26.73) | 83.94 (20.29) |
| Q5 | 67.72 (29.73) | 54.86 (29.66) | 79.84 (24.24) |
| Q6 | 50.56 (34.47) | 39.87 (30.68) | 60.63 (34.87) |
| Q7 | 53.88 (31.49) | 42.57 (28.11) | 64.54 (30.84) |
| Q8 | 74.71 (24.23) | 65.42 (24.86) | 83.46 (20.06) |
| Q9 | 54.74 (30.93) | 43.55 (27.77) | 65.3 (30.09) |
| Q10 | 69.57 (31.78) | 56.58 (33.31) | 81.81 (24.69) |
| Q11 | 74.92 (26.09) | 63.25 (27.63) | 85.91 (18.85) |
| Q12 | 67.94 (27.95) | 55.06 (27.81) | 80.07 (22.08) |

## Edge weights and variability

The edge with the greatest weight magnitude was between Q7 and Q9 (fear of re-injury and afraid of accidentally injuring knee) for all three networks (Figs 1 and 2). For the entire cohort, the Q7-Q9 association was 0.48 (95%CI [0.40 to 0.57]), whilst the associations for those who did not and did return to sports were 0.47 (95%CI [0.36 to 0.57]) and 0.48 (95%CI [0.30 to

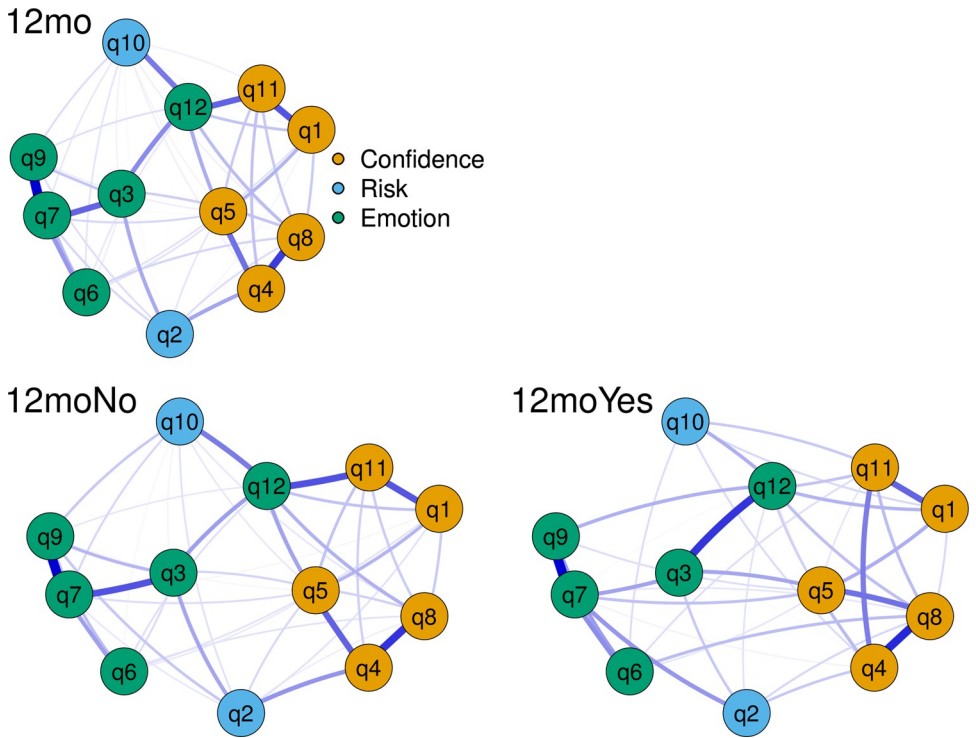

**Fig 1. Network analysis of the association between items of the ACL return to sport after injury scale.** Edges represent connections between two nodes and are interpreted as the existence of an association between two nodes, adjusted for all other nodes. Each edge in the network represents either positive regularized adjusted associations (blue edges) or negative regularized adjusted associations (red edges). The thickness and colour saturation of an edge denotes its weight (the strength of the association between two nodes). For abbreviations definition, please see Table 1 in the manuscript.

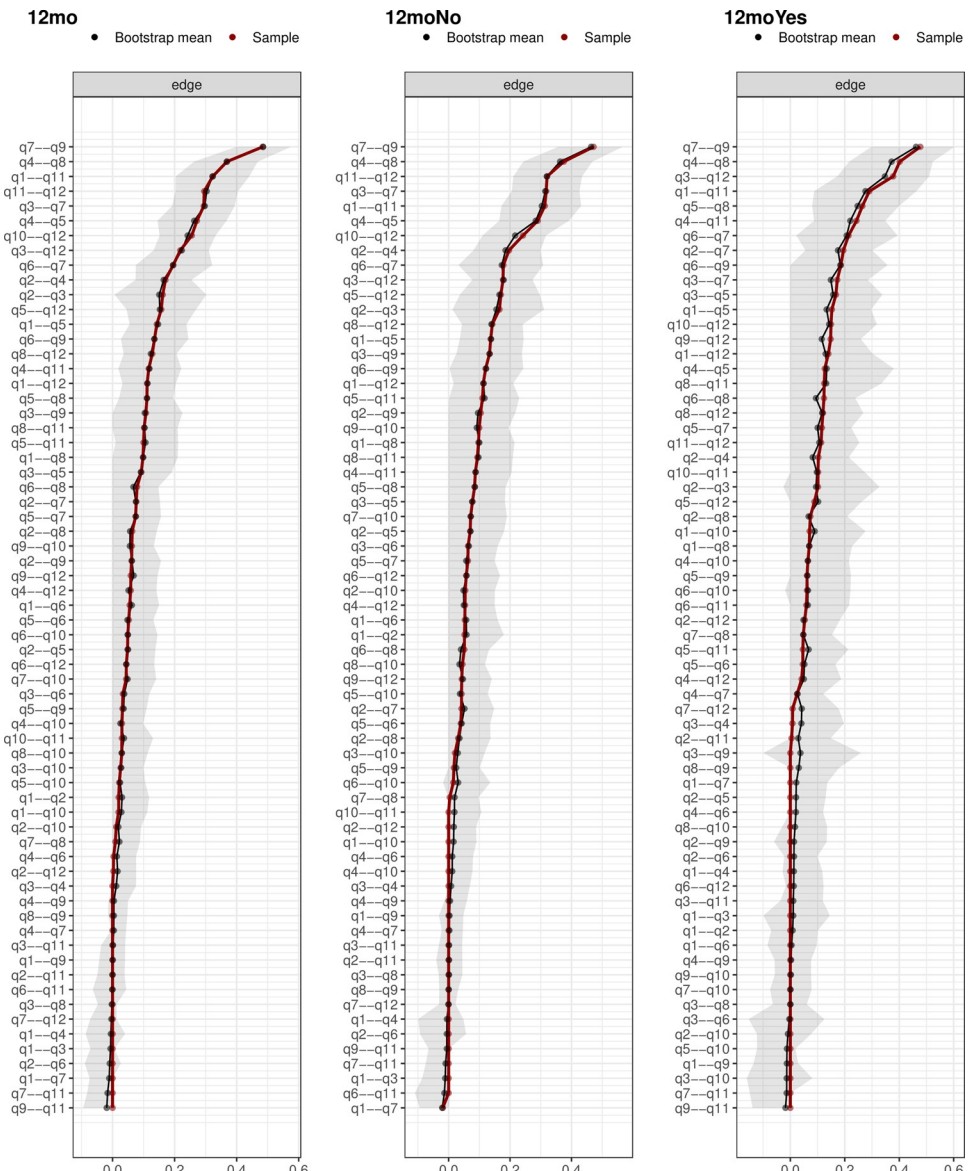

**Fig 2. Bootstrapped 95% quantile confidence interval of the estimated edge weights of the network at all follow-up time points.** "Bootstrap mean" reflects the average magnitude of edge weights across the bootstrapped samples. "Sample" reflects the magnitude of edge weights of the original network built on the entire input dataset. For abbreviations definition, please see Table 1 in the manuscript.

0.60]), respectively (Figs 1 and 2). The edge with the second greatest weight magnitude was between Q4 and Q8 (confidence in knee not giving way and confidence in knee holding up) for all three networks (Figs 1 and 2). For the entire cohort, the Q4-Q8 association was 0.37 (95%CI [0.26 to 0.47]), whilst the associations for those who did not and did return to sports were 0.38 (95%CI [0.24 to 0.47]) and 0.40 (95%CI [0.22 to 0.52]), respectively (Figs 1 and 2).

## Centrality and variability

Q12 (relaxed about sport) was the most important node for the entire cohort and the subgroup of those who did not return to sport, but the second most important node for those who did

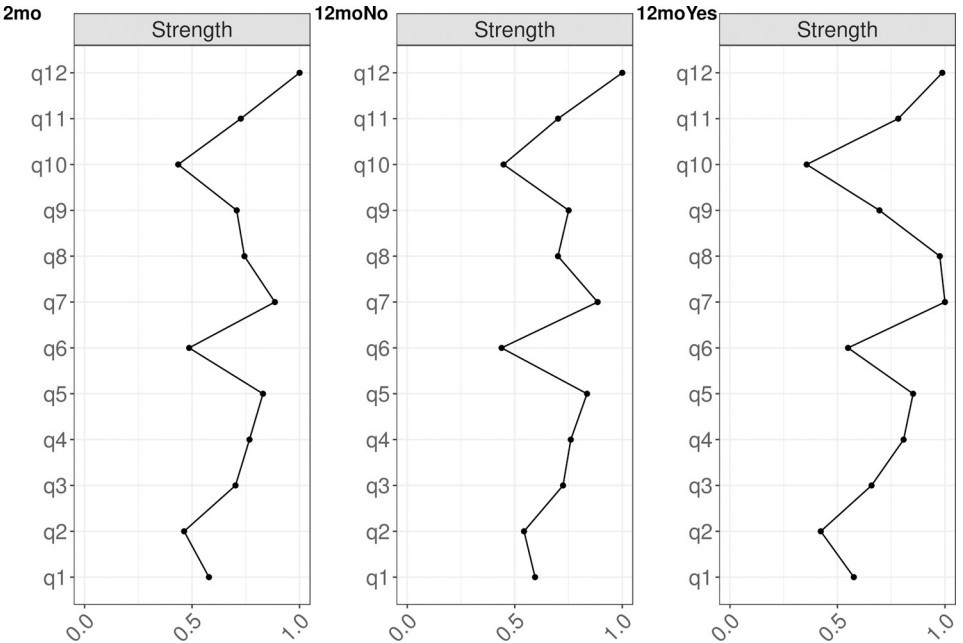

**Fig 3. Centrality measures of Closeness, Strength, and Betweenness of each node in the network at all follow-up time points.** Centrality value of 1 indicates maximal importance, and 0 indicates no importance. For abbreviations definition, please see Table 1 in the manuscript.

return to sport (Fig 3). Q7 (fear of re-injury) was the most important node for those who did return to sport, but the second most import node for the entire cohort and those who did return to sport (Fig 3). The stability of the centrality measure was 0.75, 0.75, and 0.51 for the entire cohort, those who did not return, and for those that did return to sport, respectively.

## Network comparison

Individuals who did return to sport had greater association between Q2-Q7 (likelihood and fear of reinjury, P = 0.040), Q5-Q8 (confidence in sport and confidence in knee, P = 0.044), and Q3-Q12 (nervousness and relax, P = 0.029), compared to those who did not return to sport (Fig 1). In addition, individuals who did return to sport had greater Strength centrality for Q8 (confidence in knee, P = 0.014) compared to those who did not return to sport (Fig 3).

## Discussion

Up to two-thirds of athletes may not return to their pre-injury sport level after ACLR, which could be driven in part by a perceived lack of readiness to return. The findings supported the first hypothesis, in that the items with the greatest association magnitude were those originating from the same psychological factors (*emotions*, *confidence in performance*, and *risk appraisal*) and on the whole, items from the emotions and confidence domains tended to cluster together in the network. The Centrality measure also supported our second hypothesis in that the two most important items originated from emotion-based items. Lastly, in partial support of the third hypothesis, the node which best differentiated those who did and did not return to sport was knee confidence.

This study found that fear of re-injury and how relaxed patients felt about returning to sport were the most influential items within the network for the entire cohort. However, only the knee confidence node had significantly greater influence (Strength centrality) in those that

did return to sport, compared to those that did not. Our findings are indirectly supported by the literature. One study reported that knee confidence significantly predicted lower limb function in individuals post ACL reconstruction, and that fear was not predictive after controlling for knee confidence [18]. Items reflecting confidence in the ACL-RSI could be measuring self-efficacy (SE) given that SE reflects the level of confidence to perform an activity [34]. Another study in low back pain reported that self-efficacy (a measure of confidence) was a more important mediator, than fear, in influencing the relationship between pain and disability [35]. It may be that greater levels of confidence empower an athlete to exercise control over their emotions, functioning, and events that affect the recovery of the injured knee. Network analysis cannot differentiate whether a node serves as a common cause, a common effect, or acts as a mediator. However, network analysis may serve as a highly exploratory hypothesis-generating technique to identify potential treatment targets. Hence, our findings suggest that treatments targeting knee confidence may help optimise an athlete's psychological state of readiness [36].

Modifications to the strength of associations could be just as important at influencing psychological readiness recovery in individuals with an ACL injury, as the individual or total item scores. Increases in the association magnitude between 1) likelihood and fear of reinjury (Q2-Q7), 2) confidence in sport and the knee (Q5-Q8), and 3) nervousness and relax state (Q3-Q12), differentiated those who did return from those who did not return to sport. No studies to our knowledge have reported the importance of associations between two or more variables as a determinant for returning to sport. Our findings may not come as a surprise as an athlete who has a low level of fear of re-injury but perceives the likelihood of re-injury to be high may not return to sport.

It can be argued that Q3 (nervousness about play) and Q12 (relax about play) are two opposite items reflecting the same psychological construct–hence, having high collinearity. Yet, the strengthening of association between nervousness and relaxation from those who did not return to sport to those that did, suggests that their relationship may be influenced by distinct factors. For example, athletes who returned to sport had a weakening of the association between being relaxed (Q12) and unhelpful thoughts (Q10), and also sporting confidence (Q11), compared to those who did not return (Fig 1). Also, athletes who returned to sport had a weakening of the association between being nervous (Q3) and fear of re-injury (Q7), compared to those who did not return. It may be that the weakening of some associations is important to increase psychological readiness, as having a certain level of fear may no longer negatively influence nervousness. Weakening of associations between different psychological symptoms has been thought to reflect the mechanism of change of some psychology-based treatments [37], but such mechanisms have yet to be explored in the rehabilitation of ACL injuries.

It was interesting to observe that the magnitude of association between knee-specific confidence items (Q4 and Q8) was greater than the associations between knee-specific and sports-specific confidence (e.g. between Q5 and Q8). This indicates that confidence may not generalize to all aspects of function, which ties in with Bandura observations [38], that there is "no all-purpose measure of perceived self-efficacy". This would mean that an athlete's confidence in the health of their knee might not be a good predictor of their confidence in their playing levels, and vice-versa. Greater knee-specific confidence has been shown to predict greater motor function [18], but has also been associated with a greater risk of re-injury [39,40]. This may not be surprising given that movement strategies that optimize performance also put the ACL at greater risk of injury [41]. The limitation of questionnaire-based methods to assess confidence or SE is that it is not specific to the movement demands of the athlete. Future research that looks into correlating the levels of confidence on specific athletic manoeuvres, and their

correlation with ACL loads are needed to understand the trade-offs between the movement for performance and injury risk.

A limitation of the present analysis was that this was performed without the inclusion of physical/motor and biological variables. A holistic biopsychological understanding of the mechanisms underpinning return to sports readiness will enable clinicians to better streamline their assessments and treatments to the most important factors that facilitate return to sports. It is also known that psychological responses change over time and this network analysis, therefore, represents a snapshot at one point in time over the rehabilitation period. The analysis also only included athletes who had suffered a first-time ACL injury and those with multiple ACL injuries may have different network analysis patterns. Lastly, given the exploratory nature of our between-group network comparisons, our findings require a confirmatory study.

## Conclusions

Fear of re-injury and being relaxed about playing sport were the two most important nodes in the network models that describe the return to sport readiness. Athletes who returned to sport had a greater Strength index for the variable of knee confidence, compared to those who did not return. Our findings provide candidate therapeutic targets that could inform future interventions designed to optimize return to sport rates in athletes post ACL reconstruction.

## Author Contributions

**Conceptualization:** Bernard X. W. Liew, Julian A. Feller, Kate E. Webster.

**Data curation:** Julian A. Feller, Kate E. Webster.

**Formal analysis:** Bernard X. W. Liew, Kate E. Webster.

**Funding acquisition:** Kate E. Webster.

**Investigation:** Julian A. Feller, Kate E. Webster.

**Methodology:** Kate E. Webster.

**Project administration:** Julian A. Feller, Kate E. Webster.

**Supervision:** Julian A. Feller, Kate E. Webster.

**Validation:** Bernard X. W. Liew, Kate E. Webster.

**Visualization:** Bernard X. W. Liew.

**Writing – original draft:** Bernard X. W. Liew, Julian A. Feller, Kate E. Webster.

**Writing – review & editing:** Bernard X. W. Liew, Julian A. Feller, Kate E. Webster.

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
