## [Decision Letter · Decision Letter 0]

22 Feb 2022

PONE-D-22-01655Understanding the psychological mechanisms of return to sports readiness after anterior cruciate ligament reconstructionPLOS ONE

Dear Dr. Liew,

Thank you for submitting your manuscript to PLOS ONE. After careful consideration, we feel that it has merit but does not fully meet PLOS ONE’s publication criteria as it currently stands. Therefore, we invite you to submit a revised version of the manuscript that addresses the points raised during the review process.

We look forward to receiving your revised manuscript.

Kind regards,

Richard Evans

Academic Editor

PLOS ONE

Journal Requirements:

Editor Comments:

General comments

1. Figure 1 is excellent

2. This manuscript has a nice balance of explaining the basics of Networks and presenting the new results.

Major point:

Many statistical tests were performed in this study, each with a 5% chance of being false positive (Type I error). I'm not sure how many test were performed, but at least 66 (the number of tests comparing "12-month yes" to "12-month no" edges, see lines 209+). Sixty-six tests, each with alpha=0.05 gives a studywise error rate of 96.6%. That means that there is a 96.6% chance that at least one of your statistically significant results is a Type I error.

Moreover, we would expect about 3 tests (66x0.05) to be Type I errors. That's the number of statistically significant results reported in lines 209 to 213

So, it is unlikely that your results were, in the statistical sense, shown to be more real than noise.

You have several options at this point. One would be to consult a statistician who should be knowledgable in the myriad ways of accounting for Type I error inflation.

Alternatively, treat the study as an exploratory study. Don't chose a 0.05 cutoff and don't use the words "statistically significant." Simply note that the lower p-values suggest the strongest results, and those results require a confirmatory study.

Minor points.

1. Line 64. It should read: The total score has disadvantages.

2. Table 2. There are numbers across from "Sex" (e.g., 3.24). What do they represent?

Reviewers' comments:

Reviewer's Responses to Questions

**Comments to the Author**

1. Is the manuscript technically sound, and do the data support the conclusions?

Reviewer #1: Yes

2. Has the statistical analysis been performed appropriately and rigorously? 

Reviewer #1: Yes

3. Have the authors made all data underlying the findings in their manuscript fully available?

Reviewer #1: Yes

4. Is the manuscript presented in an intelligible fashion and written in standard English?

Reviewer #1: Yes

5. Review Comments to the Author

Reviewer #1: I congratulate the authors on an excellent analysis of associations between psychological factors and return to sport after an ACL reconstruction. They used the 12-item ACL-RSI, exploring relationships between specific questions relating to confidence, emotions and risk appraisal. The main clinical implication is that treatments targeting knee confidence may need to be considered to optimise an individual’s readiness for return to sport.

The abstract is written well. I suggest that the Conclusion should be aligned more clearly with that at the end of paper. Delete the first sentence (of the abstract conclusion), and focus on the most important results.

The introduction offers relevant background, outlining the ACL-RSI and also explaining principles of network analyses – which, as a non-expert in that method, I found very useful, and will be useful for readers.

A large cohort of 441 participants were included in the study. As indicated, I am not an expert in network analysis, thus cannot comment about the analysis.

Supplementary Table: this is very useful, but should be reformatted. I suggest adding the question numbers, and providing the means (SD) in three columns for the whole cohort, those who have not returned, and those who have returned to sport respectively.

Relationships between the items are discussed clearly. An interesting findings is that there is a weakening of the association between ‘being nervous’ and ‘fear of injury’. That indicates that those who return to sport may have a fear of re-injury, but no longer feel nervous about the risk. I wonder whether that reflects the ability to manage their emotions of ‘nervousness’, while still having the rationale knowledge that there is a high risk of re-injury when returning to sport?

Task-specific confidence is discussed clearly in the paragraph starting in line 265.

Minor comments

Line 28: Replace ‘A total [of] three network analysis’ with ‘Three network analyses were used to ….’

Lines 71 and 254: ‘may be’ instead of ‘maybe’.

Line 227: delete the first ‘greater’

Line 237: I found the sentence “even though network analysis…’ difficult to follow. I suggest dividing it into 2 sentences.

Line 252: ‘reflecting’ instead of ‘refecting’

Line 289: ‘athletes’ instead of ‘athlete’

6. PLOS authors have the option to publish the peer review history of their article (what does this mean?). If published, this will include your full peer review and any attached files.

Reviewer #1: **Yes: **Gisela Sole

---

## [Author Response · Author response to Decision Letter 0]

8 Mar 2022

Please see the uploaded response to view the corrections with proper formatting.

Comments are in bold, response in normal typeset, and excerpts from manuscript are in italics.

Editor Comments

General comments

1. Figure 1 is excellent

2. This manuscript has a nice balance of explaining the basics of Networks and presenting the new results.

Reply: We thank the Editor for the positive comments, and will address all feedback below.

Major point:

Many statistical tests were performed in this study, each with a 5% chance of being false positive (Type I error). I'm not sure how many test were performed, but at least 66 (the number of tests comparing "12-month yes" to "12-month no" edges, see lines 209+). Sixty-six tests, each with alpha=0.05 gives a studywise error rate of 96.6%. That means that there is a 96.6% chance that at least one of your statistically significant results is a Type I error.

Moreover, we would expect about 3 tests (66x0.05) to be Type I errors. That's the number of statistically significant results reported in lines 209 to 213

So, it is unlikely that your results were, in the statistical sense, shown to be more real than noise.

You have several options at this point. One would be to consult a statistician who should be knowledgable in the myriad ways of accounting for Type I error inflation.

Alternatively, treat the study as an exploratory study. Don't chose a 0.05 cutoff and don't use the words "statistically significant." Simply note that the lower p-values suggest the strongest results, and those results require a confirmatory study.

Reply: We thank the Editor for this important comment. Yes, the Editor is correct. We have taken the advice of the Editor’s second recommendation, and have reworded the Methods and Discussion sections as follows:

Methods in L178

A previous study used a threshold of 0.05 to determine statistical significance during NCT [33]. Given the exploratory nature of the present study, we did not adjust this threshold for multiple comparisons. Hence, we considered between-group differences with a P value < 0.05 as having more evidence in favour of the alternative hypothesis, than differences with a P value ≥ 0.05.

Discussion in L288

Lastly, given the exploratory nature of our between-group network comparisons, our findings require a confirmatory study.

Minor points.

1. Line 64. It should read: The total score has disadvantages.

Reply: We have changed the sentence as requested.

2. Table 2. There are numbers across from "Sex" (e.g., 3.24). What do they represent?

Reply: We thank the Editor for identifying this typographical mistake. We have removed this from Table 2.

Reviewer #1

I congratulate the authors on an excellent analysis of associations between psychological factors and return to sport after an ACL reconstruction. They used the 12-item ACL-RSI, exploring relationships between specific questions relating to confidence, emotions and risk appraisal. The main clinical implication is that treatments targeting knee confidence may need to be considered to optimise an individual’s readiness for return to sport.

The abstract is written well. I suggest that the Conclusion should be aligned more clearly with that at the end of paper. Delete the first sentence (of the abstract conclusion), and focus on the most important results.

Reply: We have reworded the Abstract’s Conclusion in L 293 to read as:

Fear of re-injury and being relaxed about playing sport were the two most important nodes in the network models that describe the return to sport readiness. Athletes who returned to sport had a greater Strength index for the variable of knee confidence, compared to those who did not return. Our findings provide candidate therapeutic targets that could inform future interventions designed to optimize return to sport rates in athletes post ACL reconstruction.

The introduction offers relevant background, outlining the ACL-RSI and also explaining principles of network analyses – which, as a non-expert in that method, I found very useful, and will be useful for readers.

A large cohort of 441 participants were included in the study. As indicated, I am not an expert in network analysis, thus cannot comment about the analysis.

Supplementary Table: this is very useful, but should be reformatted. I suggest adding the question numbers, and providing the means (SD) in three columns for the whole cohort, those who have not returned, and those who have returned to sport respectively.

Reply: We have altered this Table’s format to read as:

Table 1. Anterior Cruciate Ligament Return to Sport after Injury individual item mean (1 standard deviation) scores.

Items Whole cohort (n = 441) Did not return to sport (n = 326) Return to sport (n = 115)

Q1 80.78 (22.94) 71.74 (26.36) 89.3 (14.8)

Q2 67.34 (26.02) 59.82 (25.49) 74.44 (24.53)

Q3 59.34 (31.15) 47.92 (28.53) 70.1 (29.69)

Q4 74.15 (25.68) 63.77 (26.73) 83.94 (20.29)

Q5 67.72 (29.73) 54.86 (29.66) 79.84 (24.24)

Q6 50.56 (34.47) 39.87 (30.68) 60.63 (34.87)

Q7 53.88 (31.49) 42.57 (28.11) 64.54 (30.84)

Q8 74.71 (24.23) 65.42 (24.86) 83.46 (20.06)

Q9 54.74 (30.93) 43.55 (27.77) 65.3 (30.09)

Q10 69.57 (31.78) 56.58 (33.31) 81.81 (24.69)

Q11 74.92 (26.09) 63.25 (27.63) 85.91 (18.85)

Q12 67.94 (27.95) 55.06 (27.81) 80.07 (22.08)

Relationships between the items are discussed clearly. An interesting findings is that there is a weakening of the association between ‘being nervous’ and ‘fear of injury’. That indicates that those who return to sport may have a fear of re-injury, but no longer feel nervous about the risk. I wonder whether that reflects the ability to manage their emotions of ‘nervousness’, while still having the rationale knowledge that there is a high risk of re-injury when returning to sport?

Task-specific confidence is discussed clearly in the paragraph starting in line 265.

Reply: We thank the Reviewer for this very interesting observation. Yes, we agree with the views of the Reviewer which is evidenced from Figure 1. Moving from not returning to returning to sports, the correlation between Q3 (nervousness) and Q7 (fear) reduces, but the correlation between Q7 and Q2 (risk) increases. This could mean that being in a state of nervousness is less of a determinant of fear, but that fear is driven more by risk appraisal in those who returned to sport compared to those that did not.

In our Discussion, we focused on discussing between-group differences with greater statistical evidence in support of it. With network analysis, there can be an overwhelming number of discussion points to be made. Hence, whilst the observation made by the Reviewer is interesting, we have not included it into the Revised manuscript and have instead focused on those with the greatest evidence.

Minor comments

Line 28: Replace ‘A total [of] three network analysis’ with ‘Three network analyses were used to ….’

Reply: We have reworded this sentence.

Lines 71 and 254: ‘may be’ instead of ‘maybe’.

Reply: We have changed it to “may be”.

Line 227: delete the first ‘greater’

Reply: We have delted the first “greater”.

Line 237: I found the sentence “even though network analysis…’ difficult to follow. I suggest dividing it into 2 sentences.

Reply: We have split the sentence in L239 into two to read as:

Network analysis cannot differentiate whether a node serves as a common cause, a common effect, or acts as a mediator. However, network analysis may serve as a highly exploratory hypothesis-generating technique to identify potential treatment targets.

Line 252: ‘reflecting’ instead of ‘refecting’

Reply: We have changed it to “reflecting”

Line 289: ‘athletes’ instead of ‘athlete’

Reply: We have changed it to “Athletes”

---

## [Editor Report · Decision Letter 1]

14 Mar 2022

Understanding the psychological mechanisms of return to sports readiness after anterior cruciate ligament reconstruction

PONE-D-22-01655R1

Dear Dr. Liew,

We’re pleased to inform you that your manuscript has been judged scientifically suitable for publication and will be formally accepted for publication once it meets all outstanding technical requirements.

Kind regards,

Richard Evans

Academic Editor

PLOS ONE
---

## [Editor Report · Acceptance letter]

16 Mar 2022

PONE-D-22-01655R1 

Understanding the psychological mechanisms of return to sports readiness after anterior cruciate ligament reconstruction 

Dear Dr. Liew:

I'm pleased to inform you that your manuscript has been deemed suitable for publication in PLOS ONE. Congratulations! Your manuscript is now with our production department. 

Kind regards, 

on behalf of

Dr. Richard Evans 

Academic Editor

PLOS ONE